# Study of Aging Temperature on the Thermal Compression Behaviors and Microstructure of a Novel Ni-Cr-Co-Based Superalloy

**DOI:** 10.3390/ma17143500

**Published:** 2024-07-15

**Authors:** Hualin Cai, Zhixuan Ma, Jiayi Zhang, Jinbing Hu, Liang Qi, Pu Chen, Zhijian Luo, Xingyu Zhou, Jingkun Li, Hebin Wang

**Affiliations:** School of Materials Science and Engineering, Jiangxi University of Science and Technology, Ganzhou 341000, China; 18870722342@163.com (H.C.); kaixinxiaoma6@163.com (Z.M.); 6120220276@mail.jxust.edu.cn (J.H.); qiliang@jxust.edu.cn (L.Q.); 15144712253@163.com (P.C.); 18172738364@163.com (Z.L.); cptbtptpxiu@163.com (X.Z.); 18579797472@163.com (J.L.); wanghemse@163.com (H.W.)

**Keywords:** nickel-based superalloy, thermal compression, precipitates, high-temperature mechanical property

## Abstract

Nickel-based superalloys have been widely used in the aerospace industry, and regulating the reinforcing phases is the key to improving the high-temperature strength of the alloy. In this study, a series of aging treatments (650 °C, 750 °C, 850 °C and 950 °C for 8 h) were designed to study different thermal deformation behaviors and microstructure evolutions for a novel nickel-based superalloy. Among the aged samples, the 950 °C aged sample achieved the peak stress of ~323 MPa during the thermal deformation and the highest microhardness of ~315 HV after thermal compression, which were the greatest differences compared to before deformation. In addition, the grains of the 950 °C sample exhibit deformed fibrous shapes, and the grain orientation is isotropic, while the other samples exhibited isotropy. In the 850 °C and 950 °C high-temperature aging samples, the γ′ precipitate (about 20 nm in size) is gradually precipitated, which inhibits the movement of dislocation in the grain during compression, thus inhibiting the occurrence of dynamic recrystallization and improving the high-temperature mechanical properties of the alloy.

## 1. Introduction

With the development of aeronautical technology, the thrust-to-weight ratio of aero engines has been significantly improved. In recent years, with the increasing demand of turbine discs in aero engines, the performance for nickel-based superalloy has put forward higher requirements, which need more excellent high-temperature mechanical properties, oxidation resistance, heat corrosion resistance and fatigue resistance. Nickel-based superalloys are capable of operating at temperatures in excess of 600 °C for extending long periods of time [1,2,3,4,5]. The excellent properties of precipitation-strengthened nickel-based superalloys is attributed to the presence of the γ′ phase, and the introduction of various alloying elements has resulted in the formation of strengthening precipitates γ′-Ni_3_(Al,Ti) in nickel-based superalloys [6,7,8,9,10,11]. In order to obtain a higher heat resistance of nickel-based superalloys, some elements are usually added, like Cr, Al, Ti, Nb, etc. However, these trace elements can lead to a high degree of alloying, causing severe solidification segregation and poor deformation resistance. Therefore, it is necessary to study the hot deformation resistance and microstructure evolution of nickel-based high-temperature alloys.

In recent years, there has been an increasing amount of research dedicated to investigating the thermal deformation characteristics of nickel-based superalloys [12,13,14,15,16,17,18,19]. Wan et al. [14] observed that the presence of γ′ precipitates at relatively low deformation temperatures impede the movement and rearrangement of dislocations within the original grains as well as the migration of dynamic recrystallization (DRX) grain boundaries. In addition, Wen et al. [16] observed that the impact of the duration of initial aging on the hot workability of the GH4169 Ni-based superalloy is of notable significance. Wu et al. [17] provide a concise overview of the recent progress made in understanding the coarsening characteristics of γ′ precipitates in precipitation-strengthened nickel-based superalloys. It also highlights the resulting impact on the hot deformation behavior of these superalloys, specifically emphasizing the inhibitory effect of coarse strip-like γ′ precipitates on dislocation motion during hot deformation. The DRX phenomenon in nickel-based superalloys during high-temperature deformation can lead to grain coarsening, thereby reducing the deformation resistance of the alloy [19]. Based on previous research, it is evident that the performance of DRX depends on the presence of the second phase and deformation parameters. However, there is little research on the nucleation mechanism of DRX during thermal compression.

The focus of this study is on regulating the distribution and morphology of precipitates in a novel nickel-based superalloy through a series of different aging treatments and observing the thermal compression behavior and microstructure evolution of the nickel-based superalloy through thermal compression tests and microstructure characterization. In addition, the influence of precipitates on the thermal compression behavior of the nickel-based superalloy was also revealed. This study provides guidance not only for the subsequent study of this novel superalloy but also for the development of new superalloys.

## 2. Material and Experimental Procedures

### 2.1. Materials

The ingots of the nickel-based superalloy were prepared by vacuum melting and forging after homogenized at 1150 °C for 24 h. Table 1 shows the nominal chemical compositions of this alloy, which were performed by ICP-OES (Agilent 5800 ICP-OES, Agilent Technologies, Santa Clara, CA, USA).

### 2.2. Hardness and Thermal Compression Test

The superalloy was solution treated at 1080 °C for 2 h and then cut into cylindrical specimens 15 mm in length and 10 mm in diameter. The specimens were subsequently subjected to thermal treatment at temperatures of 650, 750, 850, and 950 °C for 8 h. Subsequently, the specimens were treated by thermal compression tests with the MMS-100 thermodynamic simulation equipment (produced by Shenyang Keanjie Material Technology Co., Shenyang, China). Before the thermal compression test, it was necessary to interpose two graphite paper sheets between the specimen and the compression die to reduce friction and thereby mitigate any potential irregularities in the deformation process. Figure 1 presents the schematic diagram of the experimental procedure before and after thermal compression. Before the thermal compression, the samples were subjected to a heating procedure wherein it was heated to a temperature of 1000 °C with a rate of 10 °C/s. Subsequently, the sample is maintained at this temperature for a duration of 10 s, ensuring that all components of the sample attain the desired temperature. Subsequently, the heating process is halted, initiating the commencement of the compression phase. The compression of all specimens is conducted at a strain rate of 0.1 s^−1^ until a true strain of 50% is achieved. Subsequently, the specimens were quenched with water. The hardness was tested by an HVS-1000A Vickers hardness tester (Laizhou Huayin Experimental Instrument Co., Laizhou, China), measured with 3 kgf stress, and the testing samples after the hot compression were cut from the center along the compression axis.

### 2.3. Microstructure Observations and Mechanical Properties Test

After thermal compression, the specimens were cut along the compression axis and mechanically ground and polished; then, they were etched for 90 s with 5 g CuCl_2_ + 20 mL HCl + 5 mL CH_3_CH_2_OH and also characterized by optical microscopy (OM, performed on Axio ScopeA1 Zeiss microscope, Zeiss, Oberkochen, Germany) and scanning electron microscopy (SEM, performed on ZEISS ∑ IGMA equipment, Zeiss, Oberkochen, Germany). For the purpose of preparing the samples for electron backscatter diffraction microscope (EBSD, Zeiss, Oberkochen, Germany) observation, the samples were electrolytically etched in a solution containing 10 mL HClO_4_ and 40 mL CH_3_CH_2_OH at 19 V and for 20 s. The Oxford Aztec HKL Standard Nordlys Max2 symmetric EBSD detector mounted on a scanning electron microscope (SEM) was used for characterization, and the data were processed by channel 5 software. Low-angle grain boundaries (LAGBs, 2~10°), medium-angle grain boundaries (MAGBs, 10~15°), and high-angle grain boundaries (HAGBs, >15°) were determined based on the orientation angles of neighboring grains. Samples were mechanically ground to a 50 μm thickness for transmission electron microscopy (TEM) observations. After that, they were punched to a disc measuring 3 mm in diameter and electrolytically polished at 20 V and −25 °C in a solution containing 33% HNO_3_ and 66% CH_3_OH. On a 200 kV-operated FEI Tecnai F20 system (FEI Company, Hillsboro, OR, USA), TEM observations were made.

## 3. Results

### 3.1. Thermal Compression Deformation Behaviors

Figure 2 displays the true stress–true strain curves and peak stress values of the different aged samples. The thermal deformation behaviors of this material are significantly influenced by the different aging temperatures, as depicted in Figure 2a. Except for the 950 °C aged sample, the stress–strain curves of the other samples (650 °C, 750 °C and 850 °C) descend from true strain 0.35. However, it is worth noting that the true stress–true strain curve of the 950 °C aged sample maintains an upward trend. Figure 2b depicts the peak stress of the stress–strain curves of the aged samples. It is not difficult to discover that the peak stress value of the 950 °C aged sample achieved the highest value of 323 MPa, which is about twice that of the other samples. Thus, it can be concluded that the peak stress of the 950 °C aged sample not only does not decrease but also contributes to improving the high-temperature compression performance.

### 3.2. Hardness Variations

Figure 3 shows the hardness variations for the novel Ni-Cr-Co-based superalloy before and after thermal compression. The hardness value of the alloy tends to rise after thermal deformation, exceeding the measured value before thermal deformation. As the aging temperature increases, the microhardness also increases, from 152 to 182 HV. After thermal compression, there is a significant rise in hardness, particularly at a temperature of 950 °C, where it reaches a value of 315 HV. This finding is in accordance with the observed stress and strain fluctuations.

### 3.3. Microstructure Variations

The microstructure of the alloy after solid solution heat treatment is shown in Figure 4. The presence of massive twins and continuous distribution of the second phase in the sample, combined with the EDS results (C: 39 at.%, Fe: 36 at.% and Cr: 10 at.%), indicate that this phase is an enriched Fe-Cr carbide phase.

The microstructure evolutions of different aged samples after thermal compression were observed by EBSD. The grain orientation distribution and inverse pole diagram of the thermal-compressed aging samples are shown in Figure 5. The results imply that the samples exhibit different grain sizes and crystal preferential orientation after deformation. Significantly, the 650 °C and 850 °C aged samples mostly present equiaxed grains, and a necklace substructure (large grains surrounded by small sub-grains) is obvious, as shown in Figure 5a,c. In contrast, the 950 °C sample presents deformed fibrous grains, and the grain orientations are isotropic, as shown in Figure 5e. The blue grains are oriented with the {111} plane, the red grains indicate the {001} direction, and the green grains represent the {101} plane. Combined with Figure 5a,b, the preferential orientations of the 650 °C aged sample grains are <001>//Y0 and <101>//Z0. And in Figure 5c,d, the orientation of the 850 °C aged sample grains changes to <001>//Z0. When the equiaxed grains evolved to deformed grains (950 °C aged sample), the preferred orientations are high-density <001>//Y0 and<101>//Z0, as shown in Figure 5e,f. Actually, an important discovery reflected here is that when the aging temperature reaches 950 °C, fibrous grains are formed after thermal deformation instead of equiaxed grains, and it is speculated that certain precipitates generate inhibiting dynamic recrystallization at 950 °C, which will be discussed in the subsequent sections.

The recrystallization behaviors of the alloy after thermal deformation are revealed in Figure 6. Overall, as the aging temperature increases, the degree of recrystallization of the sample after thermal deformation gradually decreases. The recrystallization degree of Figure 6a–c is 41.6%, 29.3% and 5.4%, respectively. Figure 6 shows the typical characteristics of DRX (recrystallized grains enclose deformed grains, exhibiting a typical chain-link substructure). In Figure 6b, the increased volume fraction of substructured grains and the decreased DRX degree may be due to the inhibition of recrystallization by some precipitates. In the 950 °C aged sample, the elongated grains exhibit poor uniformity and regularity, displaying distinct features of a deformed microstructure. Additionally, only 5.4% of the recrystallized grains are observed in Figure 6c, indicating that DRX is significantly suppressed.

In order to conduct a more comprehensive analysis of the internal microstructure following thermal compression, including the examination of precipitates, dislocations, sub-grains, and other relevant features, the various samples subjected to varied ageing conditions were examined using transmission electron microscopy (TEM), as depicted in Figure 7. Figure 7a illustrates the sample aged at 650 °C following thermal compression, wherein the matrix exhibits the presence of dislocations and twins. According to the selected-area electron diffraction (SAED) pattern along [001]_γ_, no superlattice diffraction pattern of any precipitates can be found, and also no detectable precipitates can be found in TEM images. In the 850 °C aged sample, a few γ′ (platelet-like) precipitates could be observed, as shown in Figure 7b. The typical correlation SAED patterns of γ′ precipitates with zone axes parallel to [001]_γ_ are also displayed. As shown in Figure 7c, we observed plenty of γ′ (with a size about 20 nm) precipitates. These precipitates are densely distributed and entangled with dislocations. Therefore, it can be discovered that these precipitates generate at 950 °C, which should act as obstacles to impede movements of dislocations, thus improving the deformation resistance during thermal compression. 

## 4. Discussion

### 4.1. Effect of Precipitates on Thermal Compression

According to some research findings, it has been observed that many types of precipitates possess the ability to retain and immobilize residual dislocations and grain boundaries [20,21]. Based on the TEM characterizations, it is not difficult to find that the amounts of γ′ precipitates can efficiently inhibit the movements of dislocations and grain boundaries. The contact force (F) between precipitates and dislocations can be mathematically described by considering the Zener drag force and the driving force for recrystallization [22]:(1)$$F=F_{D}-F_{Z}=0.5Gb^{2}\rho_{s}+3\left(\varepsilon Gb-\gamma \right)\frac{F_{v}}{\bar{d}}$$
where $F_{D}$ is the recrystallization driving force, $F_{Z}$ is the Zener drag force provided by precipitates, $F_{v}$ is the volume fraction of precipitates, $\rho_{s}$ is the dislocation density in the Ni matrix, γ is the grain boundary energy, *G* is the shear modulus, *b* is the Burgers vector, ε is the deformation strain, and $\bar{d}$ is the average diameter of the precipitates. From the above equation, the effect of precipitates on recrystallization depends on their type, volume fraction, size and dislocation density. According to Figure 7, there are almost no observable precipitates in the 650 °C aged sample, while many γ′ precipitates can be observed in the 950 °C aged sample. Therefore, the 950 °C aged sample requires more driving force to undergo recrystallization.

Except for some empirical calculations, recent studies have reported that perturbations in precipitates have the potential to induce modifications in the energy landscape and dynamics of dislocations [23,24]. The local stacking fault energy (SFE) is facilitated by the precipitates that induce lattice distortion, resulting in dislocations movements’ inhabitation. The induced lattice strain field is believed to enhance the resistance to dislocation displacement [25]. The concept of the local lattice strain field surrounding precipitates has been proposed as a mechanism to impede the migration of dislocations. Based on Figure 7, precipitates could boost dislocation generation, and more dislocation generation is needed to allow compression to continue. By increasing the aging temperature, precipitates can nucleate faster with higher atom diffusivity, and dislocations move sluggishly with a larger drag effect. 

### 4.2. Effect of Aging on Misorientation Angle Characteristics

The increase in the DRX ratio can be ascribed to the utilization of stored energy within the grains. Moreover, the existence of precipitates might impede the progression of dislocations and the mobility of grain boundaries. Therefore, the 950 °C aged sample requires more generated energy by the thermal deformation for DRX to occur; thus, the dislocation density of the grain increases, and the alloy cannot complete the nucleation and growth of DRX. The TEM pictures (Figure 7) reveal that the dislocations in the 950 °C aged sample become entangled and proliferate significantly, resulting in a notable rise in low-angle grain boundaries (LAGBs). This can be attributed to the insufficient dissipation of the energy generated by thermal compression. 

The analysis of the microstructure’s evolution under various aging treatments is further examined through the lens of the grain boundary features. Figure 8 depicts the proportion of misorientation angles for different aging treatments after thermal compression. With the increase in aging temperature, the proportion of the LAGBs decreased from 39% to 29%, and then it increased to 61%, while the proportion of the MAGBs decreased from 3.8% to 3.3%, and then it increased to 61%, while that of the HAGBs decreased from 46% to 42%, and then it increased to 14%. Combined with the results of Figure 8, these demonstrate that different aging temperatures cause the variations of volume fractions and misorientation angles, and the notable decrease in HAGBs can be attributed to the immobilization of the grain boundary through the formation of precipitates. It is well established that recrystallized grains are predominantly encased by HAGBs. By the EBSD analysis, as depicted in Figure 6a, it can be inferred that the formation of new grains may be caused by dynamic recrystallization during thermal compression. Furthermore, it is expected that fibrous deformed grains are encompassed by LAGBs. From the Read–Shockley equation [26]:(2)$$\gamma =\gamma_{0}\theta \left(A-\mathrm{In}\theta \right);\;\gamma_{0}=\frac{Gb}{4\pi \left(1-v\right)}$$
where γ_0_ is a constant, *G* is the shear modulus, *θ* is the misorientation angle, *v* is the Poisson’s ratio, and *b* is the Burgers vector. According to this equation, it can be inferred that the dislocation angle of the atom is proportional to the energy of the grain boundary, the large dislocation angle makes the grain boundary more unstable, and the atomic motion is strengthened. Thus, the HAGBs are prone to deform, and the LAGBs have strong heat resistance in thermal compression, leading to the higher stress (as shown in Figure 2a). Based on the literature [27], there is an inherent connection between LAGBs and substructures as well as dislocations. LAGBs will undergo a transformation into MAGBs and then evolve into HAGBs as a result of the progression of DRX. Furthermore, it has been suggested that LAGBs exhibit limited mobility, as their low energy and rigid architectures make them resistant to movement [28]. Additionally, the TEM observations shown above suggest that the presence of precipitates hinders the processes of dislocation annihilation and rearrangement.

The schematic diagram illustrating the primary microstructure characteristics of the examined alloys during thermal compression is presented in Figure 9, as discussed previously. As the temperature of aging increases, there is a drop in the quantity of DRX grains, whereas LAGBs and precipitates experience an increase. In Figure 9a, the creation of these small DRX grains can be attributed to the significant plastic thermal deformation. The relaxation of interface energy and interface stress in this context serves as a potent driving factor for the merging of grains. For much higher aging temperature (Figure 9b), few precipitates appear in the matrix, and the numbers of HAGBs significantly decrease. In Figure 9c, the HAGBs almost disappear, and large amounts of γ′ precipitates are generated. The γ′ precipitated phase is distributed inside the grains as an intragranular reinforcing phase, which hinders the movement of dislocations within the grains and limits the accumulation and entanglement of dislocations, leading to the inability of the grains to form new HAGBs, suggesting that the material aged at 950 °C has excellent high-temperature strength.

## 5. Conclusions

The present study investigates the thermal compression deformation behavior and microstructure evolution of a novel Ni-Cr-Co superalloy under various aging treatments. This study also elucidates the influence of aging on the thermal compression behavior and characteristics of the misorientation angle. The findings are summarized as follows:(1)The sample after the solid solution consists of massive twins and the continuous distribution of a rich Fe-Cr carbide phase. The peak stress value of the 950 °C aged sample is 323 MPa, which is about twice that of the other samples (650, 750 and 850 °C). The hardness of the sample at 950 °C before and after thermal compression increased from 182 to 315 HV, and the improvement is relatively small in the other samples.(2)Compared with the grain form before and after thermal deformation, the 950 °C aged sample has the most significant difference. Fibrous grains with an isotropic structure are formed after thermal deformation in the 950 °C aged sample, while equiaxed grains show anisotropy in the other samples.(3)Plenty of γ′ precipitates distribute in the matrix, which pin the dislocations and sub-grain boundaries movements; thus, the 950 °C aged sample requires more driving force to undergo recrystallization, resulting in the increased volume fractions of LAGBs.

## Figures and Tables

**Figure 1 materials-17-03500-f001:**
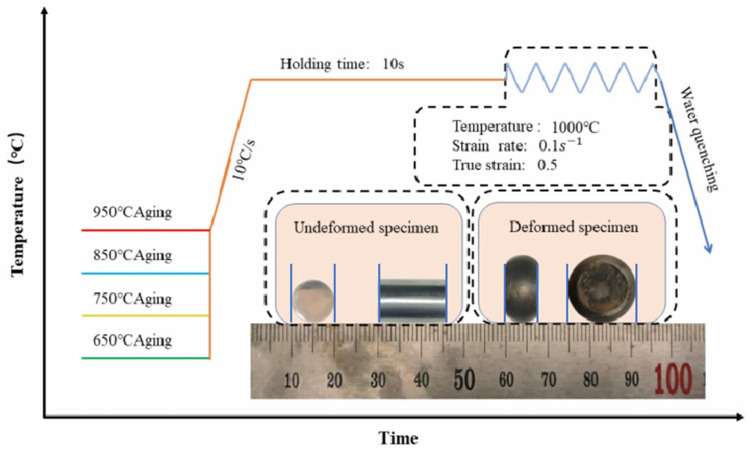
The schematic diagram experimental procedure of the thermal compression.

**Figure 2 materials-17-03500-f002:**
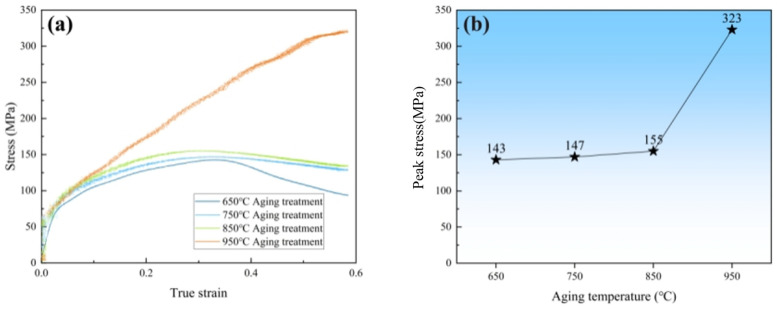
(**a**) The true stress–true strain curves and (**b**) peak stress of the aged samples.

**Figure 3 materials-17-03500-f003:**
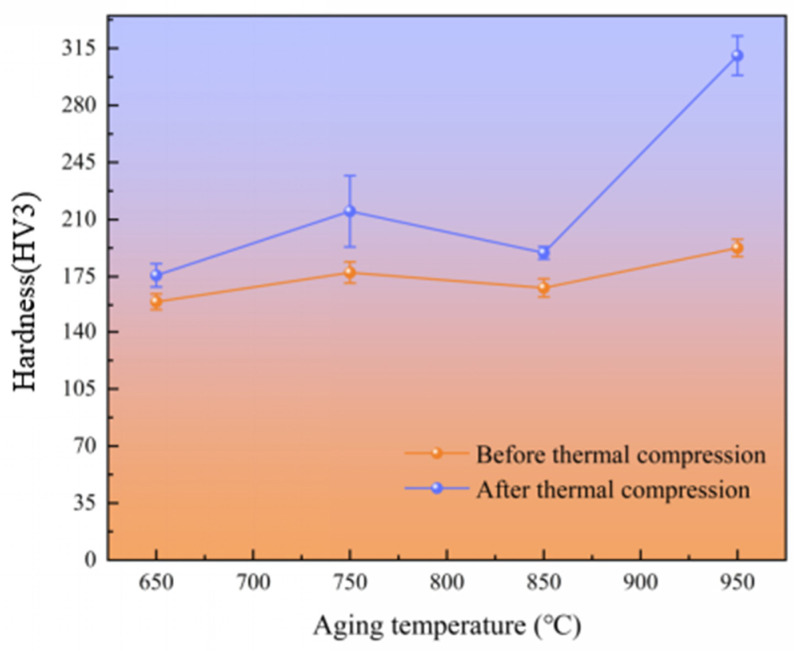
The hardness variations of the experimental alloys before and after thermal compression.

**Figure 4 materials-17-03500-f004:**
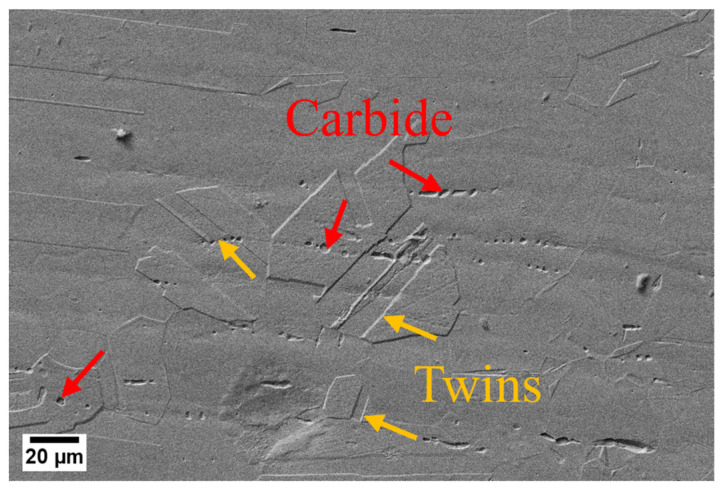
The morphology of the second phase and the twins in the solid solution of the studied samples obtained by SEM.

**Figure 5 materials-17-03500-f005:**
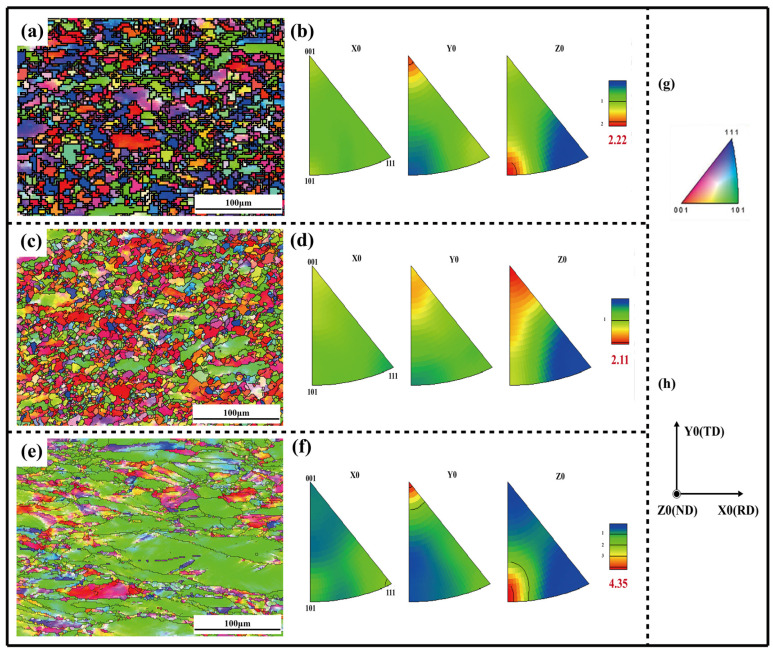
Grain orientation distributions and inverse pole figures (IPFs) of the alloy after thermal compression: (**a**,**b**) 650 °C, (**c**,**d**) 850 °C and (**e**,**f**) 950 °C aging treatment, (**g**) color key to (**a**,**c**,**e**), and (**h**) sample coordination system of (**a**,**c**,**e**).

**Figure 6 materials-17-03500-f006:**
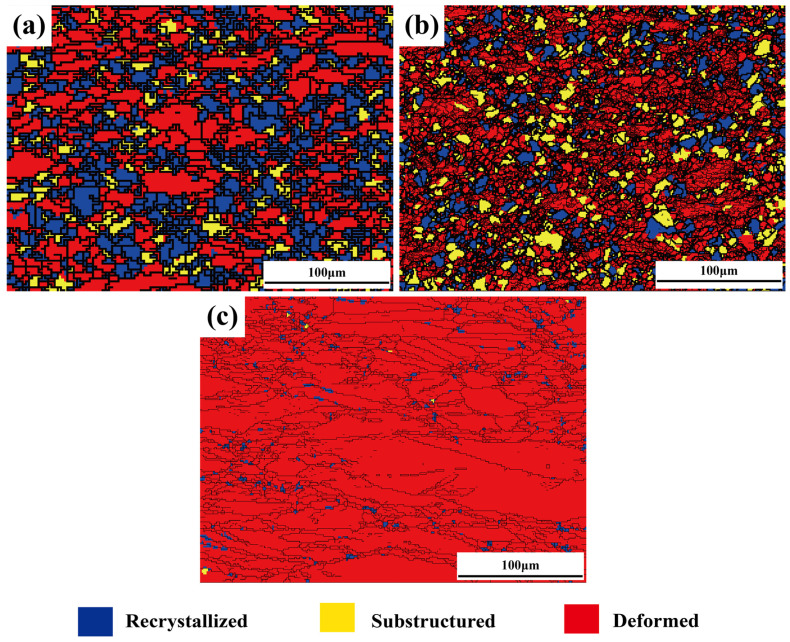
Recrystallization diagrams of the alloys after thermal compressions: (**a**) 650 °C, (**b**) 850 °C and (**c**) 950 °C aging treatment.

**Figure 7 materials-17-03500-f007:**
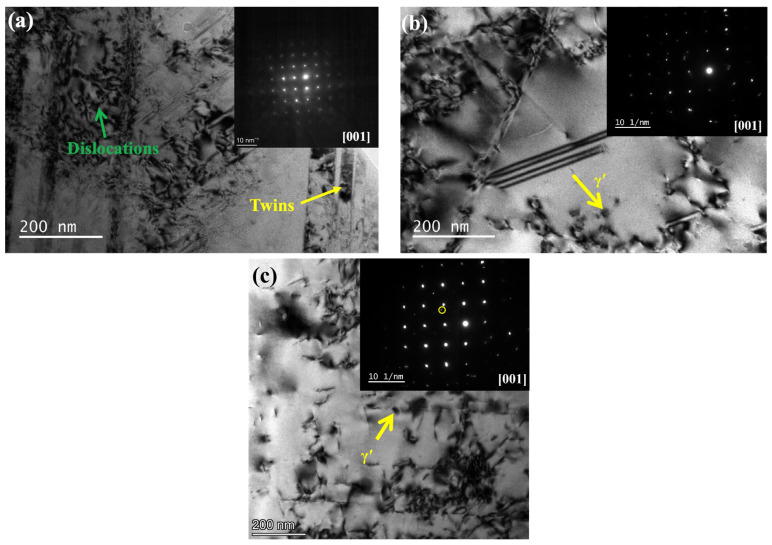
TEM images of (**a**) 650 °C, (**b**) 850 °C and (**c**) 950 °C aging alloys after thermal compression.

**Figure 8 materials-17-03500-f008:**
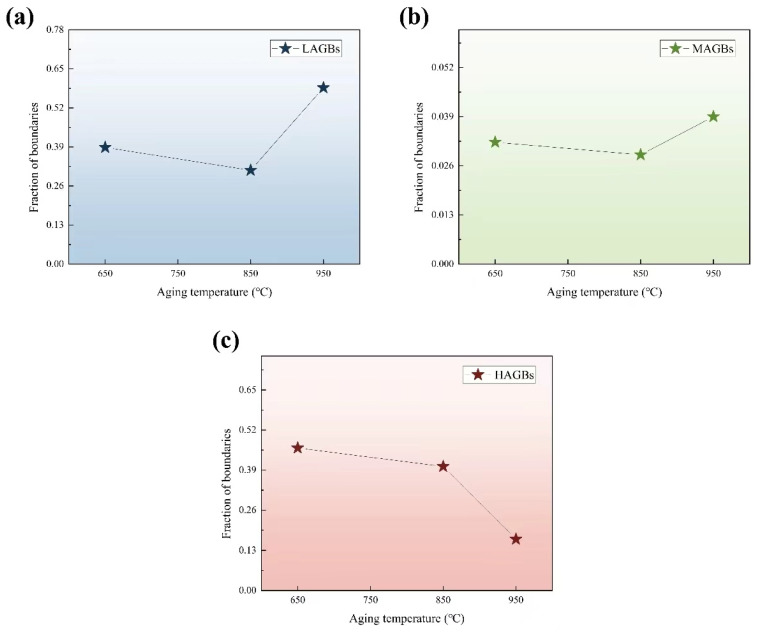
The variations of volume fraction and misorientation angles at different aging temperature after thermal compression: (**a**) LAGBs, (**b**) MAGBs and (**c**) HAGBs.

**Figure 9 materials-17-03500-f009:**
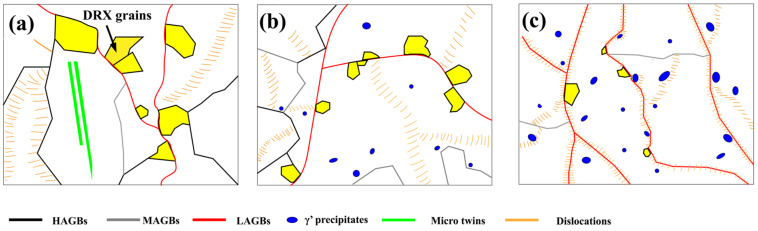
Schematic diagram for microstructure features of the different aged samples after thermal compression: (**a**) 650 °C, (**b**) 850 °C and (**c**) 950 °C aging treatments.

**Table 1 materials-17-03500-t001:** Nominal chemical composition of the nickel-based superalloy (wt.%).

Cr	Co	Ti	Al	Fe	C	B	Si	Zr	B	Cu	P	Ni
19.37	16.22	2.33	1.61	0.094	0.072	0.055	0.032	0.026	0.0055	0.005	0.0035	Bal.

## Data Availability

No data was used for the research described in the article.

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
