# Peer review of "Study of Aging Temperature on the Thermal Compression Behaviors and Microstructure of a Novel Ni-Cr-Co-Based Superalloy"

_materials, 2024, doi:10.3390/ma17143500_

Round 1

Reviewer 1 Report

Comments and Suggestions for Authors

The manuscript was revised with the title: " Study of aging temperature on the thermal compression behaviors and microstructure of a novel Ni-Cr-Co based superalloy" (materials-3062068). However, some changes must be made for publication to improve understanding and provide greater scientific support to the document. Suggestions are given below.

1. Indicate the percentage of self-citation.

2. State how the chemical composition of the nickel alloy was determined. Indicate the technique and equipment (model and brand).

3. Indicate the model and brand of all equipment and detectors used in the manuscript. Pay attention to the Microhardness Meter.

4. I suggest including a chemical analysis that shows and proves the carbides indicated in Figure 4.

5. Page 5, line 150-168: There is no clear discussion for each image in the text that precedes the discussion of Figure 5. Indicate a description for each image, such as: As seen in Fig. 5(a)...

6. Page 6, line 176-178: Indicate references that exceed the statement indicated in this text.

7. Equation 2 – I suggest increasing the size.

Comments on the Quality of English Language

Minor editing of English language required

Author Response

  1. Indicate the percentage of self-citation.

Re:The self-citation rate for this paper is 0

  1. State how the chemical composition of the nickel alloy was determined. Indicate the technique and equipment (model and brand).

Re:We used ICP-OES to determine the alloy composition, and the information about the instrument is added and marked in yellow in the text.

  1. Indicate the model and brand of all equipment and detectors used in the manuscript. Pay attention to the Microhardness Meter.

Re:We have added the manufacturer and place of manufacture of the instruments used and marked yellow.

  1. I suggest including a chemical analysis that shows and proves the carbides indicated in Figure

Re:We mentioned the EDS results for carbons in our description of Fig. 4, looking that “combined with the EDS results (C: 39 at.%, Fe: 36 at.% and Cr: 10 at.%)    ”

  1. Page 5, line 150-168: There is no clear discussion for each image in the text that precedes the

discussion of Figure 5. Indicate a description for each image, such as: As seen in Fig. 5(a)...

Re:Thank you for your suggestion, we have made the change in the text and marked yellow.

  1. Page 6, line 176-178: Indicate references that exceed the statement indicated in this text.

Re:Thank you for your suggestion, we have removed the section

  1. Equation 2 – I suggest increasing the size.

Re:Modified and marked yellow, thank you

Reviewer 2 Report

Comments and Suggestions for Authors

The authors investigated influence of aging temperature on the thermal compression and microstructure of a novel Ni-Cr-Co based superalloy. The manuscript can be published but after major revision (or eventually rejected due to a lot of things to be improved). My comments are presented below.

Abstract

-Lines 7, 9 and 11. Please uniform the style: Ni-based or nickel-based; superalloy or alloy

-English can be polished “Nickel-based superalloy has been widely used in aerospace area, various elements with 7 reinforcing effects are added to the alloy, and regulating the reinforcing phases becomes the key of 8 improving the high-temperature strength of the alloy.”

Introduction

-Lines 24-31. I don’t understand what this is. It cannot be acceptable during submission.

-Line 33. Symbol “2” should be removed.

-Line 37. “… and fatigue properties”. The fatigue properties are high, low? Please be more precise with providing the information.

-Line 38.  “Nickel-based superalloys are capable of operating at temperatures in excess of 600°C and withstand considerable stress levels for long periods of time”.

English can be polished.

- Lines 39-42

“The excellent properties of precipitation-strengthened nickel-based superalloys is attribute to the presence of the γ′ phase, and the introduction of various alloying elements has resulted in the formation of distinct primary strengthening precipitates γ′-Ni3(Al,Ti) in nickel-based superalloys [6-11].”

Please read this again, as the same information is repeated twice in one sentence. This is unacceptable. The English needs to be improved, particularly regarding the use of singular and plural forms.

-Line 43. “In order to obtain higher heat resistance of nickel-based superalloys, some reinforcing elements are usually added, like Cr, Al, Ti, Nb, etc.”

Alloying elements not reinforcing elements.

-Line 56. “…of γ' precipitates in γ'-strengthened superalloys.” This part should be re-written.

-Line 59. “..nickel based high-temperature” Pleas uniform the style: Ni-based, nickel-based or nickel based.

-Lines 65-68 “The focus of this study is to regulate the distribution and morphology of precipitates in a novel nickel-based superalloy through a series of different aging treatments, and to observe the thermal compression behavior and microstructure evolution of nickel-based superalloys through thermal compression tests and microstructure characterization.”

Did you investigate the one superalloy or more? This sentence is not clear.

-Line 68-71 “In addition, the influence of precipitates on the thermal compression behavior of nickel-based high-temperature alloys was also revealed. This study provides theoretical guidance for the study of high-temperature deformation resistance of nickel based high temperature alloys.”

Table 1 indicates that authors analysed one superalloy not superalloys.

Material and experimental procedures

- Line 75. „The ingots of the nickel-based superalloy were prepared by vacuum melting, and 75 homogenized at 1150 °C for 24 h.”

What was the size and mass of ingot?

-Table 1. Very precise content of each alloying element. It is nominal composition or measured by OES?

-Line 78. “Microhardness and Thermal compression test”.

It should be thermal not “Thermal”.

-Line 79. Alloy or Superalloy? It should be uniformed.

-Lines 82-83 “..MMS-100 thermodynamic simulation equipment.”

Journal Materials requires the more detailed information about the used devices.

-Figure 1. The quality of scalebar can be improved.

Results

-Fig. 2. How many samples were measured from each variant? The data are representative?

-Fig. 3. The 3 kgf stress during hardness test is not named as microhardness.

-Line 144. “The microstructure values of the alloy after solid solution are shown in Fig.4.”

It should be solid solution heat-treatment.

-Figure 4. Material in delivery condition was a casting or wrought product?

The experimental procedure suggests that the casting but I see on Figure the twins.

-Line 146. Did you predict the type of carbides based on one EDX measurement?

The superalloy contains 0.094% Fe and more than 2% of a strong MC carbide former, Ti. Therefore, I am not sure that Fe and Cr-rich carbides are present in the microstructure. You mentioned EDS results, but you only showed one set of values. Additionally, please check the total concentration of elements: “C: 39 at.%, Fe: 36 at.%, and Cr: 10 at.%”.

-Line 157. “The average grain sizes of the 650 °C, 850 °C and 950 °C samples are 21 μm, 8 μm and 97 μm, respectively.” How many measurements of grain size were carried out? Why does the average grain size value decrease by almost three times when the temperature increases from 650°C to 850°C? This should be clarified in more detail. Representing grain size with a histogram is more important than providing only a single value.

-Figure 7. t is quite surprising that, despite almost 4.0% γ’-formers in the chemical composition of the superalloy, you do not see superlattice reflections. In a superalloy with such a concentration of Al+Ti, the γ’ should be detectable even after solution heat treatment. During subsequent aging, the γ’ precipitates should become larger and more visible, but you need to capture the TEM images correctly, such as by using dark field. γ’ precipitates are coherent with the matrix, and the precipitates you indicate in the TEM images are not γ’.

Discussion

-Line 211. “amounts of γ′ and    precipitates”. What kind of precipitates should be there? Carbides?

-Lines 221-222. “According to the Fig.7, there are almost no observable precipitates in 650 °C aged sample, while many γ′ precipitates can be observed in 950 °C aged sample.”

You did not confirm the presence of γ′ precipitates in the microstructure. I recommend including the SAED pattern of the γ+γ′ region. Do you have any confirmation that γ′ precipitates are present?

-Line 277. Primary microstructure is used for describing the material in as-cast state.

-Figure 8. All information should be presented on one graph. Why the fraction of boundaries do not posses a standard deviation? How many locations you measured?

-Figure 9. Why the twins are present only in sample aged in 650C?

Please provide clarification why in samples aged in 650C are lack of gamma prime precipitates but in other are present?

Conclusions

-Line 299. „1) The sample after solid solution is consist with massive twins and continuous distribution rich Fe-Cr carbide phase.”

The delivery condition should be described in more detail in the Material and Experimental Procedures section. No information about plastic deformation is provided.

Line 75-76. “The ingots of the nickel-based superalloy were prepared by vacuum melting, and homogenized at 1150 °C for 24 h. Table 1 shows the nominal chemical compositions of this alloy.”

You wrote that the twins are present, but such defects can be created during plastic deformation, not casting.

-Line 299. „1) The sample after solid solution is consist with massive twins and continuous distribution rich Fe-Cr carbide phase.”

You did not confirm the presence of carbides in the microstructure. What was the aim of the TEM study if you did not analyze the type of precipitates in detail? I do not agree that the microstructure contains Fe and Cr-rich carbides.

-Line 301. The using of 3 kgf means that the study is hardness (HV3) not microhardness measurements.

-Line 303. “.,” Please improve the style of writing.

-Line 308 “(3) Plenty of γ′ precipitates distribute in the matrix”  Your SAED pattern did not show superlattice reflections.

-There is a lack of sections with information about the financial support of the research, data availability, contributions of authors, etc.

Literature

-3. References. Please check the numbering of sections

-The style of references should be improved according to the journal requirements

-Position 4, 10, 11. Lack of pages

Comments on the Quality of English Language

Moderate editing of English language required. I included my comments in the suggestions for authors. 

Author Response

Abstract

-Lines 7, 9 and 11. Please uniform the style: Ni-based or nickel-based; superalloy or alloy

Re:Modified and labelled yellow, thank you

-English can be polished “Nickel-based superalloy has been widely used in aerospace area, various elements with 7 reinforcing effects are added to the alloy, and regulating the reinforcing phases becomes the key of 8 improving the high-temperature strength of the alloy.”

 Re:We've made the changes which you suggested, and marked yellow. Thanks.

Introduction

-Lines 24-31. I don’t understand what this is. It cannot be acceptable during submission.

-Line 33. Symbol “2” should be removed.

-Line 37. “… and fatigue properties”. The fatigue properties are high, low? Please be more precise with providing the information

-Line 38.  “Nickel-based superalloys are capable of operating at temperatures in excess of 600°C and withstand considerable stress levels for long periods of time”.

English can be polished.

Re:We've made the changes you suggested and marked yellow.Thanks.

- Lines 39-42

“The excellent properties of precipitation-strengthened nickel-based superalloys is attribute to the presence of the γ′ phase, and the introduction of various alloying elements has resulted in the formation of distinct primary strengthening precipitates γ′-Ni3(Al,Ti) in nickel-based superalloys [6-11].”

Please read this again, as the same information is repeated twice in one sentence. This is unacceptable. The English needs to be improved, particularly regarding the use of singular and plural forms.

Re:Thanks to your suggestion, we've meticulously revised the passage and marked it yellow.

-Line 43. “In order to obtain higher heat resistance of nickel-based superalloys, some reinforcing elements are usually added, like Cr, Al, Ti, Nb, etc.”

Alloying elements not reinforcing elements.

Re:Modified and marked yellow.

-Line 56. “…of γ' precipitates in γ'-strengthened superalloys.” This part should be re-written.

Re:Thanks to your suggestion, we have evised the passage and marked it yellow.

-Line 59. “..nickel based high-temperature” Pleas uniform the style: Ni-based, nickel-based or nickel based.

Re:We have standardized the style of expression and marked it in yellow.

-Lines 65-68 “The focus of this study is to regulate the distribution and morphology of precipitates in a novel nickel-based superalloy through a series of different aging treatments, and to observe the thermal compression behavior and microstructure evolution of nickel-based superalloys through thermal compression tests and microstructure characterization.”

Did you investigate the one superalloy or more? This sentence is not clear.

Re:We are researching a certain superalloy, we apologize for the error in expression and have amended it.

-Line 68-71 “In addition, the influence of precipitates on the thermal compression behavior of nickel-based high-temperature alloys was also revealed. This study provides theoretical guidance for the study of high-temperature deformation resistance of nickel based high temperature alloys.”

Table 1 indicates that authors analysed one superalloy not superalloys.

Re:Thank you for your careful review and comments, revised accordingly and marked yellow.

Material and experimental procedures

- Line 75.The ingots of the nickel-based superalloy were prepared by vacuum melting, and 75 homogenized at 1150 °C for 24 h.”

What was the size and mass of ingot?

Re:The ingot is a cylinder with a diameter of 50 mm and a height of 40 mm.It is not considered to be an important parameter in this paper and is not expressed in the text.

-Table 1. Very precise content of each alloying element. It is nominal composition or measured by OES?

Re:Measured by OES, This has been added and highlighted in yellow.

-Line 78. “Microhardness and Thermal compression test”.

It should be thermal not “Thermal”.

Re:We apologise for our carelessness and have amended and yellowed the text.

-Line 79. Alloy or Superalloy? It should be uniformed.

Re:The consistency of this type of presentation has been modified and yellowed, thank you for the suggestion.

-Lines 82-83 “..MMS-100 thermodynamic simulation equipment.”

Journal Materials requires the more detailed information about the used devices.

Re:We have added information about this device and marked it yellow.

-Figure 1. The quality of scalebar can be improved.

Re:Changes have been made to figure.1.

Results

-Fig. 2. How many samples were measured from each variant? The data are representative?

Re:We made at least 3 parallel specimens for each specimen, and the low success rate of the heat distortion experiment made it more difficult to carry out the experiment.

-Fig. 3. The 3 kgf stress during hardness test is not named as microhardness.

Re:The hardness expression has been modified.

-Line 144. “The microstructure values of the alloy after solid solution are shown in Fig.4.”

It should be solid solution heat-treatment.

Re:We have made a more accurate representation and marked yellow.

-Figure 4. Material in delivery condition was a casting or wrought product?

The experimental procedure suggests that the casting but I see on Figure the twins.

Re:The material is wrought, apologize for our carelessness, now revised and marked yellow.

-Line 146. Did you predict the type of carbides based on one EDX measurement?

The superalloy contains 0.094% Fe and more than 2% of a strong MC carbide former, Ti. Therefore, I am not sure that Fe and Cr-rich carbides are present in the microstructure. You mentioned EDS results, but you only showed one set of values. Additionally, please check the total concentration of elements: “C: 39 at.%, Fe: 36 at.%, and Cr: 10 at.%”.

Re:I agree with you, this kind of carbide should be Ti rich MC type carbide, but several EDS results show Fe-Cr rich carbide, we have to respect the objective facts to describe it.

-Line 157. “The average grain sizes of the 650 °C, 850 °C and 950 °C samples are 21 μm, 8 μm and 97 μm, respectively.” How many measurements of grain size were carried out? Why does the average grain size value decrease by almost three times when the temperature increases from 650°C to 850°C? This should be clarified in more detail. Representing grain size with a histogram is more important than providing only a single value.

Re:We have deleted the paragraph because it does not fit with the preceding and following content, and the average grain size is given directly by the EBSD data processing software.

-Figure 7. t is quite surprising that, despite almost 4.0% γ’-formers in the chemical composition of the superalloy, you do not see superlattice reflections. In a superalloy with such a concentration of Al+Ti, the γ’ should be detectable even after solution heat treatment. During subsequent aging, the γ’ precipitates should become larger and more visible, but you need to capture the TEM images correctly, such as by using dark field. γ’ precipitates are coherent with the matrix, and the precipitates you indicate in the TEM images are not γ’.

Re: Firstly we are providing a nominal composition and secondly we do not agree that such an Al+Ti content would make the alloy to have a γ′ phase present after solid solution heat treatment. Diffraction spots of the γ′ phase are indeed observed in the SAED in Fig.7.

Discussion

-Line 211. “amounts of γ′ and    precipitates”. What kind of precipitates should be there? Carbides?

Re: Apologies for our carelessness, we have made the corrections and marked it yellow.

-Lines 221-222. “According to the Fig.7, there are almost no observable precipitates in 650 °C aged sample, while many γ′ precipitates can be observed in 950 °C aged sample.”

You did not confirm the presence of γ′ precipitates in the microstructure. I recommend including the SAED pattern of the γ+γ′ region. Do you have any confirmation that γ′ precipitates are present?

Re:The diffraction pattern of γ′ phase is indeed observed in the SAED in Fig.7, with the region γ + γ′.

-Line 277. Primary microstructure is used for describing the material in as-cast state.

Re:Original state of the material is wrought.

-Figure 8. All information should be presented on one graph. Why the fraction of boundaries do not possess a standard deviation? How many locations you measured?

Re:We performed one EBSD test for each set of samples, while the size-angle grain boundary occupancy ratio is derived from a large amount of data and is already statistically significant.

-Figure 9. Why the twins are present only in sample aged in 650C?

Please provide clarification why in samples aged in 650C are lack of gamma prime precipitates but in other are present?

Re:The fact that we observed the presence of twins in the TEM image of the 650°C aged compression sample but not in the TEM images of the other two samples is not an indication that there are no twins in these two samples, but is merely mentioned since the depiction of twins is not the focus of this paper.

650 ℃ aging samples due to the relatively low temperature of a large number of titanium, aluminum and other elements are difficult to desolate from the supersaturated solid solution γ-phase and form γ′-phase, whereas higher aging temperatures can promote the desolation of these elements to form γ′-phase more effectively.

Conclusions

-Line 299. 1) The sample after solid solution is consist with massive twins and continuous distribution rich Fe-Cr carbide phase.”

The delivery condition should be described in more detail in the Material and Experimental Procedures section. No information about plastic deformation is provided.

Line 75-76. “The ingots of the nickel-based superalloy were prepared by vacuum melting, and homogenized at 1150 °C for 24 h. Table 1 shows the nominal chemical compositions of this alloy.”

You wrote that the twins are present, but such defects can be created during plastic deformation, not casting.

Re:Yes, the material is in the wrought state, and we have made modifications.

-Line 299. 1) The sample after solid solution is consist with massive twins and continuous distribution rich Fe-Cr carbide phase.”

You did not confirm the presence of carbides in the microstructure. What was the aim of the TEM study if you did not analyze the type of precipitates in detail? I do not agree that the microstructure contains Fe and Cr-rich carbides.

Re:We very much agree with you, but our experimental results do indeed show Fe and Cr-rich carbides, see the two EDS spectra below.

-Line 301. The using of 3 kgf means that the study is hardness (HV3) not microhardness measurements.

Re: Changes have been made, thank you for your suggestion.

-Line 303. “.,” Please improve the style of writing.

Re: Modified. Thank you.

-Line 308 “(3) Plenty of γ′ precipitates distribute in the matrix” Your SAED pattern did not show superlattice reflections.

Re:Regarding the question about SAED, as the diffraction patterns in Fig.7(b) and (c), it is possible to see the spots of the second phase with less brightness.

-There is a lack of sections with information about the financial support of the research, data availability, contributions of authors, etc.

Re:The corresponding content has been added at the end of the article.

Literature

-3. References. Please check the numbering of sections

Re:Changes have been made.

-The style of references should be improved according to the journal requirements

Re:Modified the style of the references.

-Position 4, 10, 11. Lack of pages

Re:The corresponding missing content has been filled in, thank you for your careful reading and valuable comments!

Reviewer 3 Report

Comments and Suggestions for Authors

Dear Authors, 

I strongly recommend reading the Manuscript carefully before the new submission. 

In the introduction section, the first 9 lines are "instruction lines" and this is not acceptable in a scientific manuscript. The authors should pay more attention in the submission phase. Moreover, the quality of some images is low.

The review process is an effort-consuming process that the reviewers perform for the sake of research. On other occasions or platforms, the article would have been rejected.

Author Response

Dear reviewer,

Thank you for your careful review and comments, we have revised the article carefully.

In the introduction section, the first 9 lines are "instruction lines" and this is not acceptable in a scientific manuscript. The authors should pay more attention in the submission phase. Moreover, the quality of some images is low.

Re:We have revised the references according to the reviewer's comments and marked them in yellow.

The review process is an effort-consuming process that the reviewers perform for the sake of research. On other occasions or platforms, the article would have been rejected.

Re:We have made changes according to the reviewers' comments one by one, but we may not be able to satisfy everyone because of our lack of scientific ability or limited language skills. Please give us another chance to review the paper.

Round 2

Reviewer 2 Report

Comments and Suggestions for Authors

The work is very valuable for researchers who work with heat-resistant materials. The authors included my suggestions in the revised version. This manuscript can be published after minor revision. My suggestions are:

-Line 11. English needs to be improved.

-Line 135-136. The EDX results are presented in the wrong way. Summary concentration does not give 100%. What is the type of carbides M23C6, M3C2, or other? I you do not have a SAED pattern of carbides you can support your results by literature.

- “Fig.4 The SEM image of the studied alloy after solid solution”.

The caption should describe what is presented in Figure. I see that it is an SEM image, but you did not precise what is visible.

- Figure 7. I do not see superlattice reflections in Figure 7, which could support your discussion. If the microstructure possesses gamma prime precipitates why you do not see the superlattice dots in 001 zone axis? Are you sure that the yellow arrows indicate gamma prime, not carbides?

-Figure 8. All information should be presented on one graph.

- Fig. 3. Please indicate what is the hardness of the material in delivery condition. I can be helpful in the answer to my question in the first revision “Please provide clarification why in samples aged in 650C are lack of gamma prime precipitates but in other are present?

- Line 180. “In 850 °C aged sample, few γ′ (platelet-like) precipitates could be observed, as shown in Fig.7 (b).”

Please indicate the platelet-like γ′, because I do not see it. Arrows indicate the blocky-shaped precipitates. Please clarify if the morphology of γ′ changes with the aging temperature. It is important because if you see different types of morphology more detailed information about the size is required.

“ As  Fig.7  (c),  could  observed  plenty  of  γ′  (with  a  size  about  20  nm)  precipitates.”

20 nm is the thickness of platelet-like γ′ or the diameter of blocks? 

Comments on the Quality of English Language

I included my comment in the response to authors.

Author Response

Thank you for your suggestions, we have made adjustments and responded accordingly to your suggestions as follows:

Line 11. English needs to be improved.

Re: We've made detailed changes to the passage, thanks for the suggestion!

-Line 135-136. The EDX results are presented in the wrong way. Summary concentration does not give 100%. What is the type of carbides M23C6, M3C2, or other? I you do not have a SAED pattern of carbides you can support your results by literature.

Re: In fact the analysis of the carbide was not the focus of our study and therefore did not go into the type of that carbide. And since EDS is a semi-quantitative analysis, we only listed the elements that accounted for more to describe this second phase.

- “Fig.4 The SEM image of the studied alloy after solid solution”.

The caption should describe what is presented in Figure. I see that it is an SEM image, but you did not precise what is visible.

Re: Your suggestion is very valuable and a description has been added in the title note

- Figure 7. I do not see superlattice reflections in Figure 7, which could support your discussion. If the microstructure possesses gamma prime precipitates why you do not see the superlattice dots in 001 zone axis? Are you sure that the yellow arrows indicate gamma prime, not carbides?

Re:According to the relevant literature, the γ′ phase exists in such shapes and sizes, and since the diffraction spots are photographed only locally in a certain region, it is possible that there is no or only a small portion of this phase inside this region.

-Figure 8. All information should be presented on one graph.

Re: Thank you very much for your comments, we have considered this issue before, and finally thought that putting the three images together would look more complicated, and drawing them separately would look more concise and clear.

- Fig. 3. Please indicate what is the hardness of the material in delivery condition. I can be helpful in the answer to my question in the first revision “Please provide clarification why in samples aged in 650C are lack of gamma prime precipitates but in other are present?”

Re: We did not measure the hardness of the original samples, because the materials are subjected to solid solution and aging treatment before service, which also corresponds to the title of the article. 650 ℃ no γ′phase precipitation is because the precipitation temperature of the phase has not yet been reached, in fact, the highest service temperature of nickel-based high-temperature alloys is usually in the vicinity of the aging temperature, and thus in some cases the solid solution alloys will be used in the service, so the paper does not only investigate the effect of aging temperature on the thermal compressive properties of the materials, but also the effect of aging temperature on the hot compressive properties of the materials. Therefore, this paper not only investigates the effect of aging temperature on the thermal compression properties of the material, but also can roughly deduce the maximum service temperature of the material, and the absence of γ′phase precipitation at 650℃ also indicates that the maximum service temperature of the material is greater than 650℃.

- Line 180. “In 850 °C aged sample, few γ′ (platelet-like) precipitates could be observed, as shown in Fig.7 (b).”

Please indicate the platelet-like γ′, because I do not see it. Arrows indicate the blocky-shaped precipitates. Please clarify if the morphology of γ′ changes with the aging temperature. It is important because if you see different types of morphology more detailed information about the size is required.

“ As  Fig.7  (c),  could  observed  plenty  of  γ′  (with  a  size  about  20  nm)  precipitates.”

20 nm is the thickness of platelet-like γ′ or the diameter of blocks?

Re: Nickel-based high-temperature alloys in the γ′-phase morphology and alloy γ′-phase forming element content and heat treatment system, for example, IN 718 alloy constitutes γ′-phase element content is less, the morphology of the disc shape, and some nickel-based single-crystal high-temperature alloys as a result of the composition of the γ′-phase element content of the shape of the more will be a rectangular, and the same shape will be the time of potentiometric temperature and time and change. At the same time, the γ′ phase is not a regular disc shape, and we describe the size of the γ′ phase in three dimensions after certain statistics.

Reviewer 3 Report

Comments and Suggestions for Authors

Dear Author, 

I suggest to improve both the Abstract and the Introduction section to improve paper clarity and research objectives. 

The achived results are interesting but little alloy information is provided.

Author Response

Thank you for your suggestions.This paper investigates the effect of heat treatment on the microstructure and thermal compression properties of a novel superalloy, which not only provides a reference for the subsequent application of this alloy, but also provides a reference for the research of other novel superalloys.